# Early Growth Response 1 Contributes to Renal IR Injury by Inducing Proximal Tubular Cell Apoptosis

**DOI:** 10.3390/ijms241814295

**Published:** 2023-09-19

**Authors:** Kyuho Jeong, Jihyun Je, Theodomir Dusabimana, Hwajin Kim, Sang Won Park

**Affiliations:** 1Department of Biochemistry, Dongguk University College of Medicine, Gyeongju 38066, Republic of Korea; khjeong@dongguk.ac.kr; 2Department of Pharmacology, Institute of Medical Science, Gyeongsang National University College of Medicine, Jinju 52727, Republic of Korea; jeri1984@naver.com (J.J.); odomy2020@gmail.com (T.D.); 3Antiaging Bio Cell Factory-Regional Leading Research Center (ABC-RLRC), Gyeongsang National University, Jinju 52828, Republic of Korea; 4Department of Convergence Medical Science, Gyeongsang National University Graduate School, Jinju 52727, Republic of Korea

**Keywords:** acute kidney injury, early growth response 1, p53, proximal tubular cells, renal ischemia–reperfusion

## Abstract

Renal ischemia–reperfusion (IR) causes acute kidney injury due to oxidative stress, tubular inflammation, and apoptosis. Early growth response 1 (Egr-1) is a transcription factor belonging to the immediate early gene family and is known to regulate cell proliferation, differentiation, and survival. Egr-1 expression is induced during renal IR; however, its pathogenic role and underlying mechanisms remain elusive. Here, we investigated the function of Egr-1 during renal IR using C57BL/6 mice and cultured renal proximal tubular HK-2 cells. Egr-1 expression increased immediately, 1–4 h after IR, whereas plasma creatinine and oxidative stress increased progressively over 24 h after IR. Egr-1 overexpression showed greater increases in plasma creatinine, renal tubular injury, and apoptosis than in the control after IR. Egr-1 overexpression also showed significant neutrophil infiltration and increased pro-inflammatory cytokines (TNF-α, MIP-2, and IL-6) after IR. Consistently, proximal tubular HK-2 cells showed immediate induction of Egr-1 at 1 h after hypoxia and reoxygenation, where its downstream target, p53, was also increased. Interestingly, Egr-1 overexpression enhanced p53 levels and tubular apoptosis, while the knockdown of Egr-1 reduced p53 levels and tubular apoptosis after H_2_O_2_ treatment. Egr-1 was recruited to the p53 promoter, which activates p53 transcription, and Egr-1 induction occurred through Erk/JNK signaling kinases, as the specific inhibitors blocked its expression. Taken together, these results show that Egr-1 is upregulated in proximal tubular cells and contributes to renal IR injury by inducing tubular apoptosis, mediated by p53 transcriptional activation. Thus, Egr-1 could be a potential therapeutic target for renal IR injury.

## 1. Introduction

Acute kidney injury (AKI) is a common but serious complication in hospitalized patients, occurring in approximately 10–15% of all hospital admissions. The prevalence of AKI can be even higher in intensive care unit patients, sometimes exceeding 50% [1,2]. Despite the high incidence and mortality rates of AKI, effective medications for its prevention and treatment are still lacking. One of the primary risk factors for AKI is ischemia–reperfusion (IR), which also increases the risk of developing chronic kidney disease. IR injury leads to the disruption of renal tubular structure and function, which triggers acute inflammatory responses and resultant tubular cell death through apoptotic and necrotic pathways [3]. Ischemia causes a severe imbalance in metabolic supply and demand and leads to hypoxia-induced vascular and tubular dysfunction [4]. During reperfusion, oxidative stress increases membrane lipid peroxidation, as well as the oxidation of proteins and DNA, ultimately leading to apoptotic and necrotic cell death [5]. To develop an effective treatment that improves cell survival and recovery, it is essential to understand the underlying mechanisms involved.

Early growth response 1 (Egr-1) is a transcription factor that regulates the expression of many downstream genes involved in cell proliferation, differentiation, inflammation, and apoptosis [6]. Egr-1 is an immediate early gene that can be rapidly and transiently induced by various stressors, including extracellular pathogenic and oxidative stimuli, endogenous growth factors, cytokines and mitogens, and surgical interventions [7,8,9]. Egr-1 is expressed during kidney development, but is not detectable in normal adult kidneys [10]. However, Egr-1 is markedly induced in tubular endothelial cells during acute and chronic renal injury: acutely during the early reperfusion period following ischemic injury [11,12], and chronically during renal fibrosis and diabetic kidney injury [13,14]. Oxidative stress induces Egr-1 expression [15,16], but the precise pathogenic role of Egr-1 in renal IR injury remains unclear. Thus, characterizing the target molecules of Egr-1 and its signaling pathways is important for developing strategies to prevent renal damage following IR injury.

The transcription factor p53, which was first identified as a tumor suppressor, regulates many essential cellular functions, such as cell cycle arrest, proliferation, promoting cell senescence, and apoptosis [17]. In response to various cell stressors and DNA damage, p53 controls the transcription of target genes in key factors of cellular stress pathways. p53 has been implicated in the pathogenesis of nephrotoxicity and IR injury by regulating oxidative stress, inflammation, and tubular cell death [18,19], suggesting its critical role in renal damage and dysfunction.

In this study, we investigated the pathogenic role of Egr-1 during renal IR injury by overexpressing or silencing Egr-1. We found that Egr-1 overexpression resulted in severe renal dysfunction, due to elevated tubular inflammation and cell death after renal IR. We revealed an underlying mechanism, in which Egr-1 induction under oxidative stress significantly increased tubular apoptosis by activating p53 transcription through the Erk/JNK pathway. Thus, Egr-1 inhibition using small molecule inhibitors or silencing approaches may be promising for the treatment of renal IR injury.

## 2. Results

### 2.1. Egr-1 Expression Increased in Renal Tubules during Renal IR Injury

Plasma creatinine (Cr) was measured to evaluate renal dysfunction in mice subjected to renal IR. The Cr levels significantly increased from 4 h to 24 h of reperfusion compared to sham (Figure 1A). The reactive oxygen/nitrogen species (ROS/RNS) levels were also measured in the kidney tissues after IR. The ROS/RNS levels significantly increased after 1 h of reperfusion, and then continuously elevated until 24 h of reperfusion (Figure 1B). 

The mRNA expression of Egr-1 significantly increased (~10-fold) after 1–4 h of reperfusion and then decreased, but remained elevated until 24 h of reperfusion. The mRNA expression of p53 increased significantly at 4, 8, and 24 h after reperfusion (Figure 1C). Similar to its mRNA expression, Egr-1 protein levels increased sharply after 1 h of reperfusion, and then reduced gradually until 24 h of reperfusion (Figure 1D). Renal histological analysis showed that Egr-1 expression was not observed in sham mice, whereas it was significantly elevated at 1–4 h after IR, particularly in renal tubules (Figure 1E). To confirm the tubule-specific expression of Egr-1 after renal IR, we performed fluorescence immunostaining of Egr-1 and its respective tubular markers, aquaporin-1 (AQP1; proximal tubules) and calbindin (distal tubules). Egr-1 expression was not observed in the sham mice, but increased in the distal tubules after 1 h and 4 h of reperfusion, where calbindin was colocalized to Egr-1 (Figure 1F). Egr-1 expression was also prominent in the proximal tubules at 4 h after reperfusion, where AQP1 was colocalized to Egr-1 (Figure 1G). The results suggest a tubule-specific induction of Egr-1 and, possibly, differential responses during renal IR injury.

### 2.2. Egr-1 Increased Renal Tubular Damage and Apoptosis during Renal IR Injury

To study the role of Egr-1 during renal IR, we performed intrarenal injections of Egr-1 expression plasmid or control vector in vivo. Mice injected with the Egr-1 vector showed greater protein staining in their renal tubules (Figure 2A) and a significant induction of mRNA levels (Figure 2B) compared to mice injected with the control vector. The plasma Cr levels increased after 4 h and 24 h of reperfusion; the induction levels were much higher in Egr-1-overexpressing mice (Figure 2C). Consistently, Egr-1 overexpression showed severe tubular necrosis, medullary congestion and hemorrhage, and the development of proteinaceous casts in the kidney tissues after renal IR; Egr-1 overexpression in renal IR further increased the injury score (Figure 2D). To investigate the effect of Egr-1 overexpression on tubular apoptosis, we performed TUNEL staining and Western blot analysis of caspase-3 cleavage. After 24 h of reperfusion, apoptotic cells were significantly higher in Egr-1-overexpressing compared to those in control mice (Figure 2E). Consistently, caspase-3 cleavage was significantly induced after IR and further increased by Egr-1 overexpression. The Egr-1 downstream p53 levels were significantly increased by renal IR as well as Egr-1 overexpression, which suggests that p53-mediated tubular apoptosis may be exacerbated by Egr-1 overexpression (Figure 2F). The mRNA levels of p53 were also increased after IR and further elevated by Egr-1 overexpression (Figure 2G). These results suggest a pathogenic role of Egr-1, due to its induction of p53-mediated tubular apoptosis during renal IR injury.

### 2.3. Egr-1 Increased Tubular Neutrophil Infiltration and Cytokine Levels during Renal IR Injury

To investigate the role of Egr-1 on renal inflammation after IR, we measured the levels of neutrophil infiltration and expression of proinflammatory cytokines. Renal IR significantly increased neutrophil infiltration, which was aggravated by Egr-1 overexpression (Figure 3A). The mRNA expressions of proinflammatory cytokines (TNF-α, MIP-2, and IL-6) were significantly increased after 4 h and 24 h of reperfusion compared to those of the sham group, and these were further increased by Egr-1 overexpression (Figure 3B). The results suggest that Egr-1 promotes inflammatory responses during renal IR injury.

### 2.4. Reoxygenation Increased Egr-1 and Its Target p53 Expression in Proximal Tubular Cells

Hypoxia/reoxygenation (H/R) is known to trigger apoptosis through the p53-mediated signaling pathway in vitro. Egr-1 expression was determined after H/R in proximal tubular HK-2 cells. The mRNA levels of Egr-1 were sharply increased after 1–2 h of reoxygenation and reduced until 24 h after beginning reoxygenation (Figure 4A). Consistently, the protein levels of Egr-1 were increased after 1–2 h of reoxygenation and reduced afterwards. Interestingly, p53 expression was induced after H/R in HK-2 cells (Figure 4B). In addition, we observed the induction of Egr-1 expression after H/R in distal tubular MDCT cells; however, the p53 levels were unchanged (Appendix A). These results suggest that Egr-1 induces p53 expression, particularly in proximal tubular cells.

### 2.5. Oxidative Stress Upregulates Egr-1 and Activates p53 Transcription in Proximal Tubular Cells

To investigate the role of Egr-1 on p53-mediated tubular apoptosis under oxidative stress, HK-2 cells were exposed to hydrogen peroxide (H_2_O_2_) for 1–24 h, and the protein and mRNA levels of Egr-1 and p53 were measured. Egr-1 protein levels increased at 1–2 h after H_2_O_2_ treatment, and p53 protein levels significantly increased at 2–8 h after H_2_O_2_ treatment (Figure 5A). Egr-1 mRNA levels also increased at 1–2 h after H_2_O_2_ treatment, and p53 mRNA levels significantly increased at 16–24 h after H_2_O_2_ treatment (Figure 5B). 

We used small interfering RNA specific to Egr-1 (siEgr-1) to downregulate Egr-1 or scramble siRNA as a control in HK-2 cells. TUNEL-positive cells were induced after 24 h of H_2_O_2_ treatment in scramble-transfected cells, while this induction was blocked in siEgr-1-transfected cells (Figure 5C). To determine the effect of Egr-1 on tubular apoptosis, we analyzed the expression of apoptotic markers in cells transfected with either siEgr-1 or an Egr-1 overexpression plasmid and treated with H_2_O_2_ for 1 h and 24 h (Figure 5D). We first confirmed Egr-1 induction at 1 h, and p53 induction at 1 h and 24 h after H_2_O_2_ treatment. The levels of Egr-1 and p53 were dramatically reduced by siEgr-1, and increased by Egr-1 overexpression. Caspase-3 cleavage was significantly increased at 24 h after H_2_O_2_ treatment in scramble-transfected cells; siEgr-1 blocked caspase-3 cleavage, whereas Egr-1 overexpression aggravated the extent of caspase-3 cleavage (Figure 5D). Consistently, the mRNA levels of p53 were induced via H_2_O_2_ treatment, which was blocked by siEgr-1 but enhanced by Egr-1 overexpression (Figure 5E). Egr-1 was previously shown to promote p53 transcription, and, thus, we investigated whether Egr-1 is recruited to the p53 promoter to activate its transcription in H_2_O_2_-treated HK-2 cells using a chromatin immunoprecipitation (ChIP) assay. We found a significant enrichment of Egr-1 near the transcription start sites (TSSs) of *p53* after H_2_O_2_ treatment. We also found a recruitment of RNA polymerase II (Pol II) near the TSSs of *p53* after H_2_O_2_ treatment, while those with the IgG control antibody were not enriched (Figure 5F). The results indicate that Egr-1 promotes p53-mediated apoptosis through directly activating p53 transcription upon oxidative stress. 

### 2.6. Egr-1 Is Upregulated through Erk and JNK Pathways in Proximal Tubular Cells under Oxidative Stress

We determined the phosphorylation of Erk, p38, and JNK in the kidney tissues from the mice subjected to renal IR. The p-Erk levels increased for 1–24 h after reperfusion. The p-p38 and p-JNK levels increased shortly 1 h after reperfusion, and then reduced to their basal levels (Figure 6A). To examine the tubular activity of JNK after renal IR, we performed fluorescence immunostaining for p-JNK and AQP1. Consistently, we observed p-JNK expression in the proximal tubular cells at 1 h after reperfusion. The results suggest that JNK is activated shortly after reperfusion and required for Egr-1 induction during renal IR (Figure 6B). In addition, we determined whether Egr-1 expression is regulated through MAPKs of Erk, p38, and JNK in proximal tubular in vitro cells. We treated HK-2 cells with respective inhibitors for each kinase—U0126 for Erk, SB203580 for p38, and SP600125 for JNK—prior to 1 h of H_2_O_2_ treatment. Then, the Egr-1 expression and phosphorylation of Erk, p38, and JNK were determined using Western blot analysis. H_2_O_2_ treatment increased the protein levels of Egr-1 and the phosphorylation of Erk, p38, and JNK (Figure 7A). Erk phosphorylation (p-Erk) was inhibited by U0126, and p38 phosphorylation (p-p38) was inhibited by SB203580. JNK phosphorylation (p-JNK) was inhibited by both SP600125 and U0126. Egr-1 expression was markedly inhibited by U0126 and partially inhibited by both SB203580 and SP600125 (Figure 7A). Consistent with the Egr-1 protein, the mRNA levels showed similar responses to each inhibitor (Figure 7B). To further determine the roles of Erk, p38, and JNK on the transcription of Egr-1 under oxidative stress, we performed a luciferase assay, using a reporter construct containing the Egr-1 promoter binding sites. H_2_O_2_ treatment enhanced the transcription of Egr-1, which was completely inhibited by both U0126 and SP600125 (Figure 7C). These results indicate that Erk and JNK are essential kinases for the transcriptional activation of Egr-1 in proximal tubular cells upon oxidative stress. 

## 3. Discussion

Egr-1 has been shown to be induced at the early reperfusion phase after ischemic injury in differential gene expression studies, and it plays a crucial role in tissue damage and organ dysfunction [11,12,20,21]. Egr-1 induction under oxidative stress and its downstream signaling have been studied with regard to diseased conditions [9,22,23,24]; however, not many studies have been conducted on Egr-1 regulation in renal IR injury, and the underlying molecular mechanism is not clearly understood. In this study, we first found an induction of Egr-1 at 1–4 h after renal IR, and Egr-1 overexpression exacerbated renal dysfunction, tubular inflammation, and apoptosis in mice subjected to renal IR. Egr-1 overexpression in vitro in proximal tubular cells upregulated p53 under oxidative stress, while Egr-1 knockdown reduced tubular apoptosis, as well as the p53 levels induced by oxidative stress. We revealed that Egr-1 is directly bound to the p53 promoter, and activated p53 transcription and tubular apoptosis as a molecular mechanism. In addition, Egr-1 induction occurred through Erk and JNK pathways. Taken together, these results showed an important role of Egr-1 on p53-mediated tubular apoptosis under oxidative stress, which suggests that Egr-1 regulation has the therapeutic potential to alleviate renal IR injury.

During renal IR injury, tubular cell death occurs via apoptosis and necrosis. Ischemia induces severe ATP depletion, resulting in intracellular calcium accumulation and protease activation, leading to necrosis [25]. During reperfusion, apoptosis occurs due to the mitochondrial release of apoptotic factors and caspase activation [26]. Proximal tubular cells are highly susceptible to apoptosis, possibly through the action of stress kinases upstream of apoptotic factors [5]. The administration of growth factors or caspase inhibitors ameliorates IR injury by inhibiting tubular apoptosis and suppressing inflammatory responses [27]. The transcription factor p53 plays an important pathogenic role during renal IR injury, as the pharmacological or hormonal inhibition of p53 reduces the effects of renal IR injury [17,18]. Particularly, the proximal tubule cell-specific deletion of p53 reduces apoptosis, inflammation, and interstitial fibrosis after both short- and long-term renal IR [19]. In addition, the excess production of ROSs is a critical factor in early reperfusion injury, and ischemic preconditioning or suppression of ROS production alleviates renal injury and could be a treatment option for AKI [28,29,30]. However, the interplay between p53 and ROSs in renal tubular apoptosis during IR injury remains unclear. Egr-1 is known to be a redox-sensitive transcription factor, and it is rapidly induced upon exposure to oxidants. Although Egr-1 mediates cell survival and proliferation as a protective mechanism against oxidant injury, H_2_O_2_-induced Egr-1 expression and the subsequent apoptotic response are direct consequences of oxidant damage during IR [9]. Here, we observed enhanced ROS production after renal IR, and a subsequent increase in Egr-1 expression in vivo. We found that Egr-1 bound to the p53 promoter, activating its transcription and increasing p53-mediated apoptosis. Overexpression of Egr-1 exacerbated renal injury, as shown in the increased neutrophil infiltration, cytokine production, and apoptotic marker induction. These results support that Egr-1 induction directly activates p53 expression and tubular damage during IR.

Interestingly, Egr-1 has shown tubule-specific expression during early reperfusion in vivo, and p53 induction was found particularly in proximal tubular cells in vitro. Egr-1 expression increased at 1–4 h after reoxygenation, and subsequent p53 upregulation occurred at 8–24 h after reoxygenation in proximal tubular cells; however, p53 levels were unchanged in distal tubular cells. Distal tubules exhibit greater resistance to hypoxia, ischemia, and oxidative injury than proximal tubules during renal IR, thus maintaining their integrity [31]. The relative resistance of distal tubules to ischemic injury may be due to their increased expression of anti-apoptotic Bcl-2 and Bcl-X(L) proteins [32,33], where regenerative growth factors synergistically increase cell survival [33]. P53 regulates the transcription of pro-apoptotic Bcl-2 family members and may also impact Bcl-2 activity [34,35,36]. These observations suggest that p53 upregulation at early reperfusion is critical for inducing tubular apoptosis in proximal tubules, contributing to severe renal damage after IR. 

P53 is not only a proapoptotic factor, but also a regulator of other processes in AKI, such as cell cycle progression and senescence. Cellular senescence after acute stress is gradually induced, and p21, a downstream of p53, is upregulated at 2 h after renal IR and increased for 3 days [37]. Expression of the cyclin-dependent kinase inhibitors, p21 and p16, normally induces senescent growth arrest and promotes senescence-associated secretory phenotypes and degenerative pathologies [38,39]. Further investigation may reveal a possible role of Egr-1 on p53-mediated senescence during renal recovery after IR. 

Egr-1 expression is upregulated in the kidneys with tubulointerstitial nephritis (TIN), and Egr-1-deficient mice are protected from TIN through reduced renal inflammation and fibrosis [13]. Egr-1 expression is also elevated in renal failure patients, and persistent expression of Egr-1 is shown to aggravate nephrotic progression in many chronic kidney diseases [40,41,42]. Consistently, our preliminary data showed that Egr-1 expression after renal IR was 4- to 6-fold higher for 14 days than it was in sham mice. We suggest that early and transient induction of Egr-1 promotes p-53-mediated tubular apoptosis to eliminate severely damaged cells. In contrast, long-term low but sustained Egr-1 induction leads to renal inflammation and fibrosis, accelerating the AKI-CKD transition. However, a recent study showed that early and transient induction of Egr-1 enhances kidney repair by activating tubular SOX9 expression 3 days after renal IR [43]. The conflicting roles of Egr-1 in the early and chronic stages of AKI may be related to maladaptive control of repair mechanisms in AKI through continuous stimulation of differential Egr-1 downstream targets. 

The present study has some limitations. Firstly, we did not monitor the long-term effects of Egr-1 after renal IR. Secondly, our study did not examine Egr-1 deficiency in tubular-specific mechanisms in vivo. Future studies will endeavor to illustrate these issues and reveal effective therapeutic strategies by manipulating Egr-1. Current therapeutic options for AKI are limited to renal dialysis or transplantation, and, thus, Egr-1 and its downstream have potential therapeutic benefits in alleviating renal tubular injury and preventing renal failure in patients.

## 4. Materials and Methods

### 4.1. Animals 

Male C57BL/6 mice (7 weeks old) were obtained from Koatech (Pyeongtaek, South Korea) and maintained in the animal facility at Gyeongsang National University. All animal experiments were approved by the Institutional Board of Animal Research at Gyeongsang National University (GNU-171130-M0058), and conducted in accordance with the National Institutes of Health guidelines for laboratory animal care. Mice were maintained with a 12 h light/dark cycle and provided freely with water and standard chow.

### 4.2. Mouse Model of Renal IR Injury

Mice were habituated for 1 week and randomly divided into groups (n = 4–8). The mice were anesthetized with zoletil (0.5 mg/kg; Virbac Laboratories, Carros, France), and placed supine on a heating pad under a heat lamp to maintain a consistent body temperature. After abdominal incision, microvascular clamps were placed on both renal pedicles to block flow in the renal artery and vein for 25 min. Internal organs were kept hydrated with warm saline during the ischemic period, and the incision was sutured after removal of the clamps. The sham model operation was performed without clamping. Mice were sacrificed at the indicated times after reperfusion, and blood and kidney tissues were collected. Kidney tissues were rapidly frozen in liquid nitrogen for storage at −80 °C, or fixed in 10% buffered formalin. Blood was obtained from an inferior vena cava by using a heparinized syringe and centrifuged at 3000× *g* for 20 min. Plasma creatinine was measured using Pure Auto S CRE-N (Daiichi Sankyo, Tokyo, Japan).

### 4.3. Egr-1 Overexpression In Vivo

Mouse Egr-1 cDNA (NM_007913) was subcloned into a pcDNA3.1(+) mammalian plasmid vector. Egr-1 or control empty vector (100 μg) was mixed with 50 μL of transfection reagent NANOPARTICLE (Altogen Biosystems, Las Vegas, NV, USA) in a total volume of 100 μL. Mice were anesthetized, kidneys were exposed, and the mixture was directly injected into the renal capsules of both kidneys. The mice were subjected to renal IR 24 h after the suturing of the abdomen.

### 4.4. Cell Culture and Treatment 

Human proximal tubular HK-2 cells (CRL-2190) and mouse distal tubular MDCT cells (CRL-3250) were purchased from ATCC and grown in DMEM/F12 medium (1:1 mixture of Dulbecco’s modified Eagle medium and Ham’s F-12 medium), supplemented with 10% (*v*/*v*) fetal bovine serum (Hyclone Laboratories, Logan, UT, USA) and 1% penicillin/streptomycin in a 5% CO_2_ incubator at 37 °C. To induce a hypoxic condition, cells were incubated in 1% O_2_ and 5% CO_2_ at 37 °C for 24 h using a water-jacketed hypoxia chamber (Astec Co., Kasuya, Fukuoka, Japan), and oxygen was replaced with a 5% CO_2_ incubator at 37 °C. The cells were cultured in a serum-free medium during hypoxia. Cells were treated with 600 μM of H_2_O_2_ (Sigma-Aldrich, St. Louis, MO, USA) to stimulate oxidative stress. Cells were treated with U0126 (Sigma-Aldrich), SB203580 (Sigma-Aldrich), and SP600125 (Sigma-Aldrich) as indicated.

### 4.5. Silencing and Overexpression of Egr-1 in HK-2 Cells

Small interfering RNAs (siRNAs) against scramble or human Egr-1 (siEgr-1) were purchased from Bioneer (Daejeon, Republic of Korea). Cells were transfected with siRNA using Lipofectamine RNAiMAX (Invitrogen, Carlsbad, CA, USA) and incubated for 24 h. For Egr-1 overexpression, cells were transfected with 1 μg of Egr-1 plasmid using a Lipofectamine LTX transfection reagent (Invitrogen), according to the manufacturer’s instructions, and incubated for 24 h. Egr-1 expression was validated with PCR and Western blot analysis.

### 4.6. Total ROS/RNS Assay

Total ROS/RNS levels in kidney tissues were quantified using an OxiSelect ROS/RNS assay kit (STA-347, Cell Biolabs, San Diego, CA, USA), following the manufacturer’s protocol. Kidney tissue lysates were prepared by homogenizing them in RIPA buffer, followed by sonication and centrifugation at 12,000× *g* for 15 min at 4 °C. Fluorescence intensity was measured using a Tecan Infinite F200 PRO microplate reader (Tecan Austria GmbH, Grödig, Austria).

### 4.7. Hematoxylin and Eosin (H&E) Staining 

Kidney tissues were fixed in 10% formalin for 24 h, processed for paraffin embedding, and sectioned at 5 µm. The sections were stained with H&E (Sigma-Aldrich), using a standard protocol, and imaged using a CKX41 light microscope (Olympus, Tokyo, Japan). Renal injury scores were determined semi-quantitatively as previously described [44].

### 4.8. Terminal Deoxynucleotidyl Transferase dUTP Nick-End Labeling (TUNEL) Assay 

To determine apoptotic cell death of kidney tissues and HK-2 cells, a TUNEL assay was performed using an in situ cell death detection kit (Roche, Mannheim, Germany) according to the manufacturer’s instructions. Images were captured using a CKX41 light microscope (Olympus), and the number of TUNEL-positive cells was counted from five microscopic fields (200×) per section from each group.

### 4.9. Immunohistochemistry (IHC)

The fixed tissue sections were deparaffinized, rehydrated, and antigen-retrieved in sodium citrate buffer (10 mM, pH 6.0) for 20 min. Endogenous peroxidase activity was blocked with 0.3% hydrogen peroxide, and nonspecific binding was blocked with 10% normal horse serum. The sections were incubated with a primary antibody against Ly-6B.2 antibody (Bio-Rad, Hercules, CA, USA) or Egr-1 antibody (Cell Signaling Technology, Danvers, MA, USA) overnight at 4 °C, and then with a biotinylated secondary antibody (Vector Laboratories, Burlingame, CA, USA) for 1 h at room temperature. The sections were incubated in an avidin–biotin–peroxidase complex solution (ABC solution; Vector Laboratories) for 30 min, and developed using a 3,3′-diaminobenzidine (DAB) Peroxidase Substrate Kit (Vector Laboratories). Then, the sections were counterstained with hematoxylin, and the images were obtained using a CKX41 light microscope (Olympus).

### 4.10. Immunofluorescence Staining

Sections were blocked with 2.5% normal horse serum and incubated overnight at 4 °C in primary antibodies as follows: Egr-1 and phosphorylated c-Jun N-terminal kinase (p-JNK) from Cell Signaling, aquaporin-1 (AQP1) from Santa Cruz Biotechnology (Dallas, TX, USA), and Calbindin from Abcam (Cambridge, MA, USA). After washing, the sections were incubated with Alexa Fluor488-conjugated and/or Alexa Fluor594-conjugated secondary antibodies (Vector Laboratories, Burlingame, CA, USA). Fluorescence was visualized using a Fluoview 1000 IX-81 confocal microscope (Olympus). The fluorescence intensity of p-JNK was analyzed using Image J software, Version 1.52a (NIH, Bethesda, MD, USA).

### 4.11. Western Blot Analysis

Kidney tissues and HK-2 or MDCT cells were homogenized in ice-cold RIPA buffer with protease inhibitors (Thermo Fisher Scientific, Waltham, MA, USA), sonicated, and incubated for 20 min on ice. The lysates were separated and transferred to polyvinylidene difluoride membranes. After blocking in 5% skim milk or 3% bovine serum albumin for 1 h at room temperature, the membranes were incubated overnight at 4 °C in primary antibodies as follows: Egr-1, p53, extracellular signal-regulated kinase (Erk), phosphorylated Erk (p-Erk), JNK, phosphorylated JNK (p-JNK), p38, phosphorylated p38 (p-p38), caspase-3 (Cell Signaling Technology), and β-actin (Sigma-Aldrich). After washing, the membranes were incubated with the appropriate horseradish peroxidase-conjugated secondary antibodies (Bio-Rad) for 1 h at room temperature. Then, the membranes were developed using a Clarity™ Western ECL Substrate (Bio-Rad), and the protein band intensity was analyzed using the ChemiDoc XRS+ System (Bio-Rad). The list of antibodies used in this study is shown in Appendix A.

### 4.12. Real Time-Quantitative PCR (RT-qPCR)

Total RNA was extracted with Trizol (Invitrogen, Carlsbad, CA, USA) and converted to cDNA using the RevertAid Reverse Transcription System (Thermo Fisher Scientific) according to the manufacturer’s instructions. Real-time PCR analysis was performed with a CFX Connect real-time PCR System using iQ SYBR Green Supermix (Bio-Rad). PCR was performed with an initial preincubation step for 10 min at 95 °C, followed by 45 cycles of 95 °C for 10 s, annealing at 57 °C for 10 s, and extension at 72 °C for 10 s. The relative mRNA levels were normalized to those of glyceraldehyde 3-phosphate dehydrogenase (GAPDH). The list of primer sequences is shown in Appendix A.

### 4.13. Chromatin Immunoprecipitation Assay (ChIP Assay) 

The chromatin immunoprecipitation (ChIP) assay was performed as described previously [45]. HK-2 cells were fixed with 1% formaldehyde for 10 min at RT and quenched in 125 mM glycine for 10 min. Cells were incubated for 15 min at 4 °C with a lysis buffer (5 mM HEPES pH 8.0, 85 mM KCl, 0.5% NP-40, and protease inhibitors); nuclear pellets were collected via centrifugation at 3000× *g* for 5 min and resuspended in the RIPA buffer. Crude nuclear lysates were sonicated into 200 to 500 bp fragments with the Ultrasonic Disintegrator Sonicator S-4000 (QSonica, Newton, CT, USA). DNA fragments from each sample were incubated overnight at 4 °C with antibodies of Egr-1, RNA polymerase II, or control IgG with Protein G Dynabeads (Thermo Fisher Scientific). Chromatin immunoprecipitants were washed, and DNA was eluted via reversed crosslinking using Proteinase K (Sigma-Aldrich). DNA was purified using a QIAquick PCR purification kit (Qiagen, Hilden, Germany). Samples were analyzed with qPCR with iTaq Universal SYBR Green Supermix (Bio-Rad) using the CFX Connect system (Bio-Rad). The qPCR data were normalized based on the amount of input chromatin. The list of ChIP primers is shown in Appendix A.

### 4.14. Dual-Luciferase Reporter Assay

HK-2 cells were transfected using a Lipofectamine LTX reagent (Invitrogen) with a luciferase reporter plasmid, pGL2, either containing the Egr-1 promoter (Egr-1-Luc) or empty (Promega, Madison, WI, USA). After 24 h of transfection, the cells were treated with H_2_O_2_ for 1 h, and luciferase activity was determined with the Dual-Luciferase Reporter Assay system (Promega, Madison, WI, USA). The gene encoding the Renilla luciferase (RLuc) (Promega) was transfected simultaneously as an internal control.

### 4.15. Statistical Analysis

Statistical difference was assessed using one-way analysis of variance (ANOVA), followed by Tukey’s post hoc multiple comparisons tests for multiple groups. All values are expressed as means ± standard error of mean (SEM). A *p* value < 0.05 was considered as statistically significant.

## Figures and Tables

**Figure 1 ijms-24-14295-f001:**
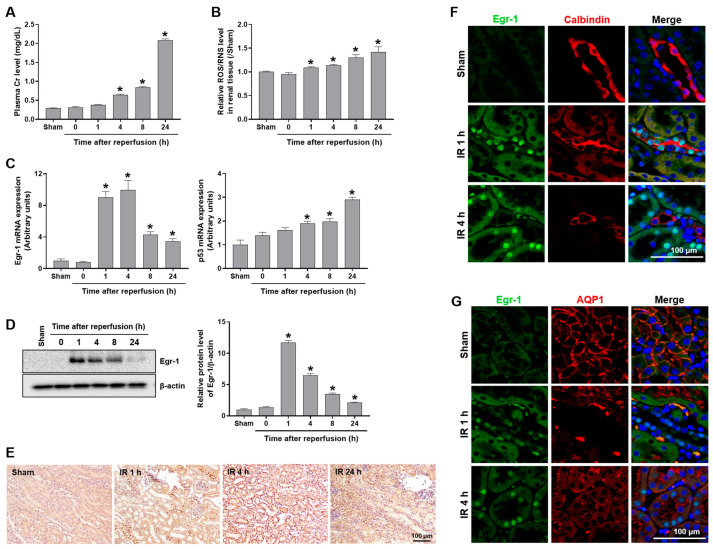
Egr-1 expression induced after renal IR injury in mice. Plasma and kidney tissues were collected from C57BL/6 mice subjected to either a sham operation, or 25 min of bilateral renal ischemia, followed by reperfusion, for the indicated durations. (**A**) Plasma creatinine and (**B**) renal ROS/RNS levels were measured. (**C**) Egr-1 and p53 mRNA expression were measured using real-time quantitative PCR analysis. (**D**) Egr-1 protein expression was measured using Western blot analysis; relative protein levels were determined with β-actin as a loading control. (**E**) Representative images of Egr-1 immunohistochemical staining: Egr-1 was localized in renal tubules. (**F**,**G**) Representative images of Egr-1 immunofluorescence staining: Egr-1 was co-stained with a proximal tubular marker (AQP1) or a distal tubular marker (calbindin). The data are presented as mean ± SEM. * *p* < 0.05 vs. sham group. Scale bar, 100 μm. Egr-1—early growth response 1; IR—ischemia–reperfusion; ROS/RNS—reactive oxygen/nitrogen species; AQP1—aquaporin-1.

**Figure 2 ijms-24-14295-f002:**
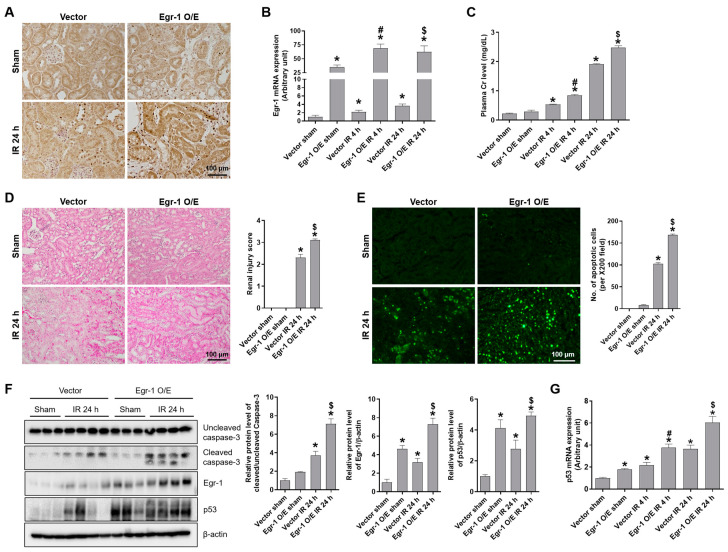
Egr-1 overexpression exacerbated renal damage and tubular apoptosis after renal IR injury in mice. At 24 h after intrarenal injection of Egr-1 plasmid or control vector, C57BL/6 mice were subjected to either a sham operation, or 25 min of bilateral renal ischemia, followed by 4 or 24 h of reperfusion. Kidney tissues were collected at the indicated times. (**A**) Representative images of Egr-1 immunohistochemical staining. (**B**) Egr-1 mRNA expression was measured using real-time quantitative PCR analysis. (**C**) Plasma creatinine levels were measured. (**D**) Representative H&E images of the kidney: the extent of renal injury was scored. (**E**) Representative TUNEL images: the number of TUNEL-positive cells was counted. (**F**) The protein expressions of Egr-1, p53, and uncleaved or cleaved caspase-3 were measured using Western blot analysis; relative protein levels were determined with β-actin as a loading control. (**G**) p53 mRNA expression was measured using real-time quantitative PCR analysis. The values are expressed as mean ± SEM. * *p* < 0.05 vs. control sham group. ^#^ *p* < 0.05 vs. control IR 4 h group. ^$^ *p* < 0.05 vs. control IR 24 h group. Scale bar, 100 μm. Egr-1—early growth response 1; IR—ischemia–reperfusion; H&E—hematoxylin and eosin; TUNEL—terminal deoxynucleotidyl transferase dUTP nick-end labeling.

**Figure 3 ijms-24-14295-f003:**
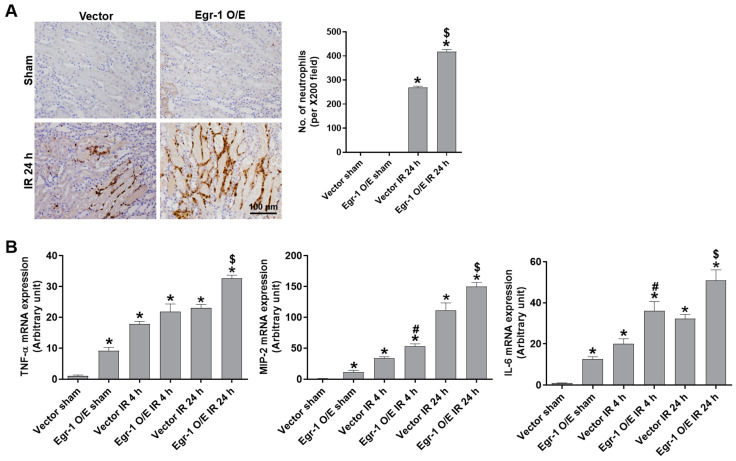
Egr-1 overexpression exacerbated neutrophil infiltration and proinflammatory cytokine production after renal IR injury in mice. At 24 h after intrarenal injection of Egr-1 plasmid or control vector, C57BL/6 mice were subjected to either a sham operation, or 25 min of bilateral renal ischemia, followed by 4 or 24 h of reperfusion. Kidney tissues were collected at the indicated times. (**A**) Representative images of Ly-6B.2 immunohistochemical staining for neutrophil infiltration: the number of stained neutrophils was counted. (**B**) The mRNA expressions of TNF-α, MIP-2, and IL-6 were measured using real-time quantitative PCR analysis. The values are expressed as mean ± SEM. * *p* < 0.05 vs. control sham group. ^#^ *p* < 0.05 vs. control IR 4 h group. ^$^ *p* < 0.05 vs. control IR 24 h group. Scale bar, 100 μm. Egr-1—early growth response 1; IR—ischemia–reperfusion; TNF-α—tumor necrosis factor-α; MIP-2—macrophage inflammatory protein-2; IL-6—interleukin-6.

**Figure 4 ijms-24-14295-f004:**
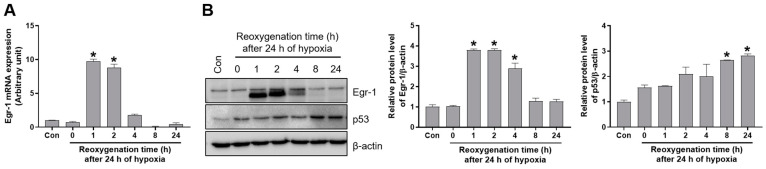
Egr-1 and p53 expression were induced by hypoxia and reoxygenation in renal proximal tubular cells. Proximal tubular HK-2 cells were exposed to 24 h of hypoxia, followed by reoxygenation at the indicated times. (**A**) Egr-1 mRNA expression was measured using real-time quantitative PCR analysis (**B**) The protein expressions of Egr-1 and p53 were measured using Western blot analysis; relative protein levels were determined with β-actin as a loading control. The values are expressed as mean ± SEM. * *p* < 0.05 vs. control group. Egr-1—early growth response 1; HK-2—human kidney 2.

**Figure 5 ijms-24-14295-f005:**
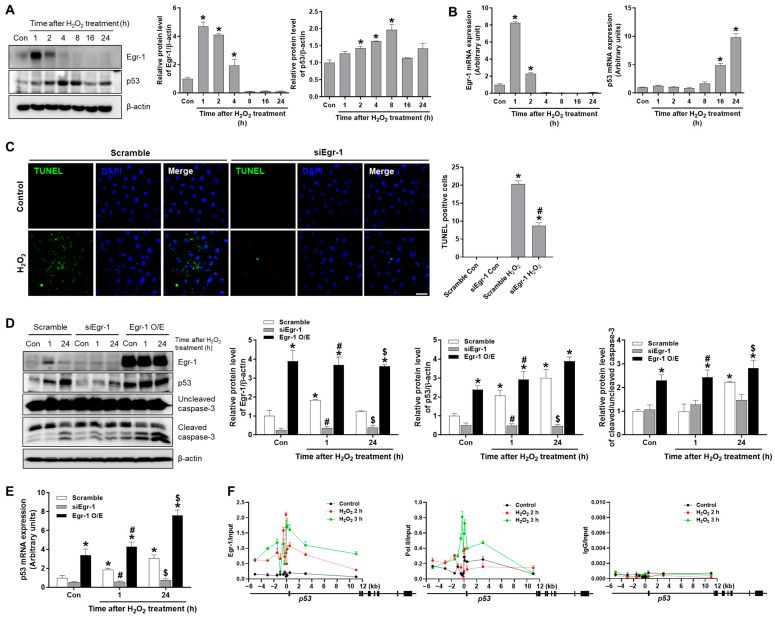
Egr-1 knockdown reduced apoptosis and p53 levels in H_2_O_2_-treated proximal tubular HK-2 cells. HK-2 cells were treated with H_2_O_2_ (600 μM) for the indicated durations. (**A**) The protein expressions of Egr-1 and p53 were measured using Western blot analysis; relative protein levels were determined with β-actin as a loading control. (**B**) Egr-1 and p53 mRNA expression were measured using real-time quantitative PCR analysis. (**C**) Cells were transfected with either scramble or Egr-1 siRNA 24 h prior to H_2_O_2_ treatment and incubated with H_2_O_2_ for 24 h. Representative TUNEL images are shown: the number of TUNEL-positive cells was counted. (**D**,**E**) Cells were transfected with either scramble, Egr-1 siRNA, or Egr-1 overexpression vector 24 h prior to H_2_O_2_ treatment, and then incubated with H_2_O_2_ for 1 or 24 h. (**D**) The protein expressions of Egr-1, p53, and uncleaved or cleaved caspase-3 were measured using Western blot analysis; relative protein levels were determined with β-actin as a loading control. (**E**) p53 mRNA expression was measured using real-time quantitative PCR analysis. (**F**) Cells were treated with H_2_O_2_ for 2 or 3 h and subjected to ChIP assay with either anti-Egr-1, anti-RNA polymerase II, or anti-IgG as a control, around the TSSs of the *p53* gene. The values are expressed as mean ± SEM. * *p* < 0.05 vs. control or scramble control group. ^#^ *p* < 0.05 vs. scramble H_2_O_2_ 1 h group. ^$^ *p* < 0.05 vs. scramble H_2_O_2_ 24 h group. Scale bar, 100 μm. Egr-1—early growth response 1; HK-2—human kidney 2; TUNEL—terminal deoxynucleotidyl transferase dUTP nick-end labeling; ChIP—chromatin immunoprecipitation; TSSs—transcription start sites.

**Figure 6 ijms-24-14295-f006:**
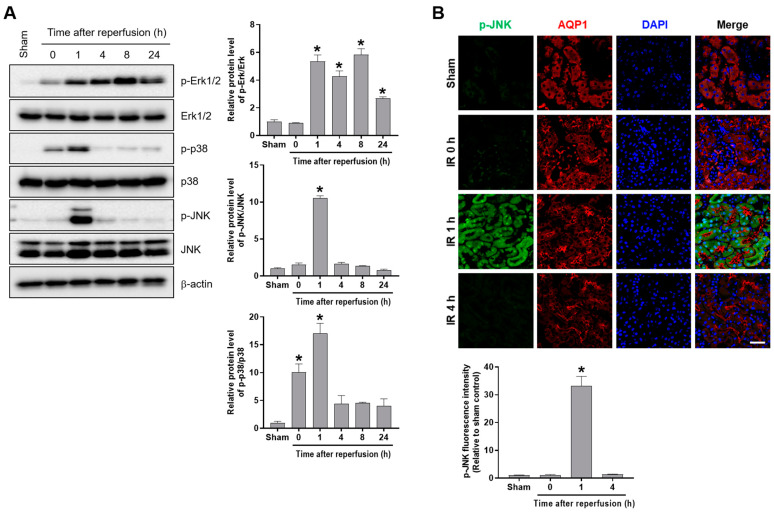
Renal IR increased MAPK phosphorylation, and p-JNK was induced in proximal tubules at 1 h after reperfusion of mice. Kidney tissues were collected from C57BL/6 mice subjected to either a sham operation, or 25 min of bilateral renal ischemia, followed by reperfusion of the indicated durations. (**A**) The protein expressions of p-Erk1/2, Erk1/2, p-p38, p38, p-JNK, and JNK were measured using Western blot analysis; relative protein levels were determined with β-actin as a loading control. (**B**) Representative images of p-JNK immunofluorescence staining: Egr-1 was co-stained with a proximal tubular marker, AQP1. The fluorescence intensity of p-JNK was measured using Image J. The data are presented as mean ± SEM. * *p* < 0.05 vs. sham group. Scale bar, 100 μm. Egr-1—early growth response 1; IR: ischemia–reperfusion; MAMK—mitogen-activated protein kinase; JNK—c-Jun N-terminal kinase; Erk—extracellular signal-regulated kinase; AQP1—aquaporin-1.

**Figure 7 ijms-24-14295-f007:**
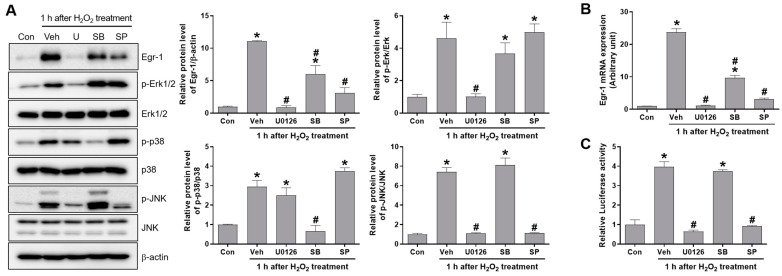
Egr-1 expression was induced through the phosphorylation of Erk and JNK in H_2_O_2_-treated proximal tubular cells. HK-2 cells were pretreated with either U0126 (100 nM), SB203580 (10 nM), or SP600125 (10 μM) 1 h prior to H_2_O_2_ (600 μM) treatment, and then the cells were collected after 1 h of H_2_O_2_ treatment. (**A**) The protein expressions of Egr-1, p-Erk1/2, Erk1/2, p-p38, p38, p-JNK, and JNK were measured using Western blot analysis; relative protein levels were determined with β-actin as a loading control. (**B**) Egr-1 mRNA expression was measured using real-time quantitative PCR analysis. (**C**) Dual-luciferase activity was measured in the cells transfected with reporter constructs, containing Egr-1 binding sites with luciferase (Egr-1-Luc), or Renilla luciferase (RLuc) as an internal control. The values are expressed as mean ± SEM. * *p* < 0.05 vs. control group. ^#^ *p* < 0.05 vs. vehicle+ H_2_O_2_-treated group. Egr-1—early growth response 1; JNK—c-Jun N-terminal kinase; Erk—extracellular signal-regulated kinase.

## Data Availability

The data presented in this study are available upon request from the corresponding author.

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
