# Peer review of "Early Growth Response 1 Contributes to Renal IR Injury by Inducing Proximal Tubular Cell Apoptosis"

_ijms, 2023, doi:10.3390/ijms241814295_

Round 1
Reviewer 1 Report
This article investigated the role of the transcription factor Early Growth Response-1 (Egr-1) in renal ischemia-reperfusion (IR) injury in mice and cultured kidney tubule cells. The authors found that Egr-1 expression increased in kidney tubules after IR injury in mice. Overexpression of Egr-1 exacerbated kidney dysfunction, inflammation, and apoptosis after IR compared to control mice. In cultured proximal tubule cells, Egr-1 expression also increased after hypoxia-reoxygenation treatment to mimic IR injury.
The authors revealed a mechanism where Egr-1 activates transcription of the p53 gene, leading to increased tubule cell apoptosis under oxidative stress. Egr-1 bound directly to the p53 promoter and activated its transcription. Knockdown of Egr-1 reduced p53 levels and apoptosis, while Egr-1 overexpression increased p53 and apoptosis in tubule cells. Egr-1 induction occurred through the Erk and JNK signaling pathways. These results suggest Egr-1 plays a pathogenic role in tubule apoptosis via p53 regulation after kidney IR injury, and targeting Egr-1 could have therapeutic potential.
Comments:
- The study only examined the short-term effects of Egr-1 overexpression after IR injury in mice. Looking at longer-term impacts on kidney function and structure could reveal additional roles for Egr-1.
- The in vivo model overexpressed Egr-1 globally rather than just in kidney tubule cells. A kidney-specific or tubule-specific Egr-1 knockout mouse would better define its renal effects.
- The signaling pathways were only examined in cultured tubule cells. Confirming the roles of Erk/JNK in Egr-1 regulation in vivo is needed.
- Only male mice were used. Testing female mice would reveal any sex-specific differences.
- The study did not look at impacts of suppressing Egr-1 expression/activity after IR injury. This could support Egr-1 as a therapeutic target.
- Additional Egr-1 target genes beyond p53 may be involved and warrant investigation.
- Testing if known Egr-1 inhibitors reduce tubule apoptosis after IR would further validate Egr-1 as a target.
- Only proximal tubule cells were used for in vitro experiments. Comparing results to distal tubules could explain differences.
- The clinical relevance is unclear. Testing human kidney cells and samples could better translate the findings.
- The mechanisms involved in Egr-1 induction after IR are still not fully defined. More detailed signaling studies could elucidate this.
Author Response
The responses to the reviewer's comments include pictures, so we attach them as a file.

Reviewer 2 Report
AKI represents a pathology which often associates high-risk of morbididity and mortality and therefore, it is important to establish an early diagnosis, to determine the risk factors and consequently, to be able to initiate the adequate form of therapy. Recently, Egr-1 has been highlighted as a potential marker for a rapid detection of renal injury, as several studies confirmed that Egr-1 is induced in tubular endothelial cells during acute and chronic renal impairment. Nevertheless, the exact pathogenic role of this factor is a matter of debate, therefore, the present study may clarify different aspects related to Egr-1 correlation with renal injury onset. The methodology and results were clearly explained and the conclusions were supported by the findings. I congratulate you for your research, having only few minor suggestions:
- lines 81-86 ("Mice were subjected to renal ischemia … levels in the kidney tissues after IR") describe the methodology, so please include this paragraph in the Methodology section.
- better include Figure 8 in the Introduction.
- please include a list of the used acronyms below each figure.
Author Response

(The authors gave the same response as above.)
